# Prediction of the development of delirium after transcatheter aortic valve implantation using preoperative brain perfusion SPECT

**Masashi Takeuchi[1,2☯], Hideaki Suzuki[1,3☯], Yasuharu Matsumoto[4], Yoku Kikuchi[1], Kentaro Takanami[5], Toshihiro Wagatsuma[6], Jun Sugisawa[1], Satoshi Tsuchiya[1], Kensuke Nishimiya[1], Kiyotaka Hao[1], Shigeo Godo[1], Tomohiko Shindo[1], Takashi Shiroto[1], Jun Takahashi[1], Kiichiro Kumagai[7], Masahiro Kohzuki[2,8], Kei Takase[5], Yoshikatsu Saiki[7], Satoshi Yasuda[1,9‡*], Hiroaki Shimokawa[1,10‡]**

**1** Department of Cardiovascular Medicine, Tohoku University Graduate School of Medicine, Sendai, Japan, **2** Department of Rehabilitation Medicine, Tohoku University Hospital, Sendai, Japan, **3** Division of Brain Sciences, Imperial College London, London, United Kingdom, **4** Department of Cardiology, Tagawa Hospital, Tagawa, Japan, **5** Department of Diagnostic Radiology, Tohoku University Graduate School of Medicine, Sendai, Japan, **6** Department of Anesthesiology and Perioperative Medicine, Tohoku University Graduate School of Medicine, Sendai, Japan, **7** Department of Cardiovascular Surgery, Tohoku University Graduate School of Medicine, Sendai, Japan, **8** Department of Internal Medicine and Rehabilitation Science, Tohoku University Graduate School of Medicine, Sendai, Japan, **9** National Cerebral and Cardiovascular Center, Suita, Japan, **10** Graduate School of Medicine, International University of Health and Welfare, Narita, Japan

☯ These authors contributed equally to this work.
‡ SY and HS also contributed equally to this work.
* syasuda@cardio.med.tohoku.ac.jp

## Abstract

### Objectives

Delirium is an important prognostic factor in postoperative patients undergoing cardiovascular surgery and intervention, including transcatheter aortic valve implantation (TAVI). However, delirium after transcatheter aortic valve implantation (DAT) is difficult to predict and its pathophysiology is still unclear. We aimed to investigate whether preoperative cerebral blood flow (CBF) is associated with DAT and, if so, whether CBF measurement is useful for predicting DAT.

### Methods

We evaluated CBF in 50 consecutive patients before TAVI (84.7±4.5 yrs., 36 females) using $^{99m}$Tc ethyl cysteinate dimer single-photon emission computed tomography. Preoperative CBF of the DAT group (N = 12) was compared with that of the non-DAT group (N = 38) using whole brain voxel-wise analysis with SPM12 and region of interest-based analysis with the easy-Z score imaging system. Multivariable logistic regression analysis with the presence of DAT was used to create its prediction model.

### Results

The whole brain analysis showed that preoperative CBF in the insula was lower in the DAT than in the non-DAT group (P<0.05, family-wise error correction). Decrease extent ratio in

Tohoku University Graduate School of Medicine (med-kenkyo@grp.tohoku.ac.jp)(No. 2018-1-567), the participants in this study were given written-informed consents, in which their data are not planned to be open to the public or to be shared with other researchers.

**Funding:** This work was supported by the Grants-in-Aid program from the Japan Society for the Promotion of Science (20K07776). The funder had no role in study design, data collection and analysis, decision to publish, or preparation of the manuscript.

**Competing interests:** The authors have declared that no competing interests exist.

the insula of the DAT group (17.6±11.5%) was also greater relative to that of the non-DAT group (7.0±11.3%) in the region of interest-based analysis (P = 0.007). A model that included preoperative CBF in the insula and conventional indicators (frailty index, short physical performance battery and mini-mental state examination) showed the best predictive power for DAT (AUC 0.882).

## Conclusions

These results suggest that preoperative CBF in the insula is associated with DAT and may be useful for its prediction.

## Introduction

Aortic stenosis (AS) is the most common valvular heart disease in the elderly worldwide, leading eventually to angina, syncope, heart failure and sudden death along with gradual progression of valve calcification [1, 2]. As transcatheter aortic valve implantation (TAVI) is a well-established minimally invasive treatment, its target group is expected to be even older and high-risk patients with multiple diseases [3–5]. Delirium after TAVI (DAT) occurs especially in elderly patients [6] and is associated with poor long-term prognosis and increase in healthcare cost [7]. The prevalence of DAT is up to 44.0%, which is higher than that of postoperative delirium in surgical aortic valve replacement (SAVR) and other surgeries [8]. However, the pathophysiology of DAT is still unclear. Moreover, DAT is difficult to predict because of its multifactorial nature [9–11] and consequently tends to be missed in clinical practice [12, 13]. Thus, a useful approach to predict DAT remains to be developed for patients undergoing TAVI [14].

A possible risk of delirium is acute brain dysfunction due to cerebral circulatory failure [15]. Cognitive impairment is prevalent in patients with severe AS and its improvement is noted after TAVI in some cases [16–18]. We previously demonstrated that TAVI increases regional cerebral blood flow (CBF), which was more prominent in patients with increased cardiac output than in those without it [18]. Cognitive impairment is associated with reduced CBF [18, 19] and delirium after surgery [20]. Thus, AS-associated cerebral hypoperfusion may be a neural substrate and risk factor for DAT. We previously demonstrated that brain perfusion single-photon emission computed tomography (SPECT) is a useful imaging technique to evaluate regional CBF in patients with cardiovascular disease [18, 21]. With respect to delirium, SPECT studies have been conducted after the onset, without reports of SPECT assessments being conducted before the onset [22, 23].

In the present study, we thus tested our hypothesis that preoperative CBF is associated with the development of DAT after TAVI in elderly patients with severe AS using brain perfusion SPECT imaging and, if so, whether preoperative CBF is useful for the prediction of DAT.

## Materials and methods

### Patient enrollment

From January 2017 to February 2020, we enrolled consecutive 97 patients with severe AS who were candidates for TAVI in our Tohoku University Hospital. Severe AS was defined as an aortic-valve area of less than 0.8 cm$^2$, a mean aortic-valve gradient of 40 mmHg or more, or a peak aortic-jet velocity of 4.0 m per second or more [4]. Before an acquisition of brain SPECT, 36 patients were excluded for the following reasons; emergent surgery (N = 2), New York Heart Association (NYHA) functional class [22] IV (N = 12), severe carotid artery stenosis

(N = 2), psychiatric disorders (N = 3), cerebral aneurysm (N = 2), preoperative delirium (N = 1) and unavailability of brain SPECT scanning before TAVI (N = 14). After TAVI, 11 patients were excluded for the following reasons; postoperative heart failure and infection (N = 4), reintubation (N = 1), pacemaker insertion (N = 4) and postoperative hemorrhage (N = 2). Finally, 50 patients were included in the present study (Fig 1).

The local heart team, including cardiologists, cardiac surgeons and anesthesiologists, determined the indication and approach of TAVI procedure and the type of transcatheter valve

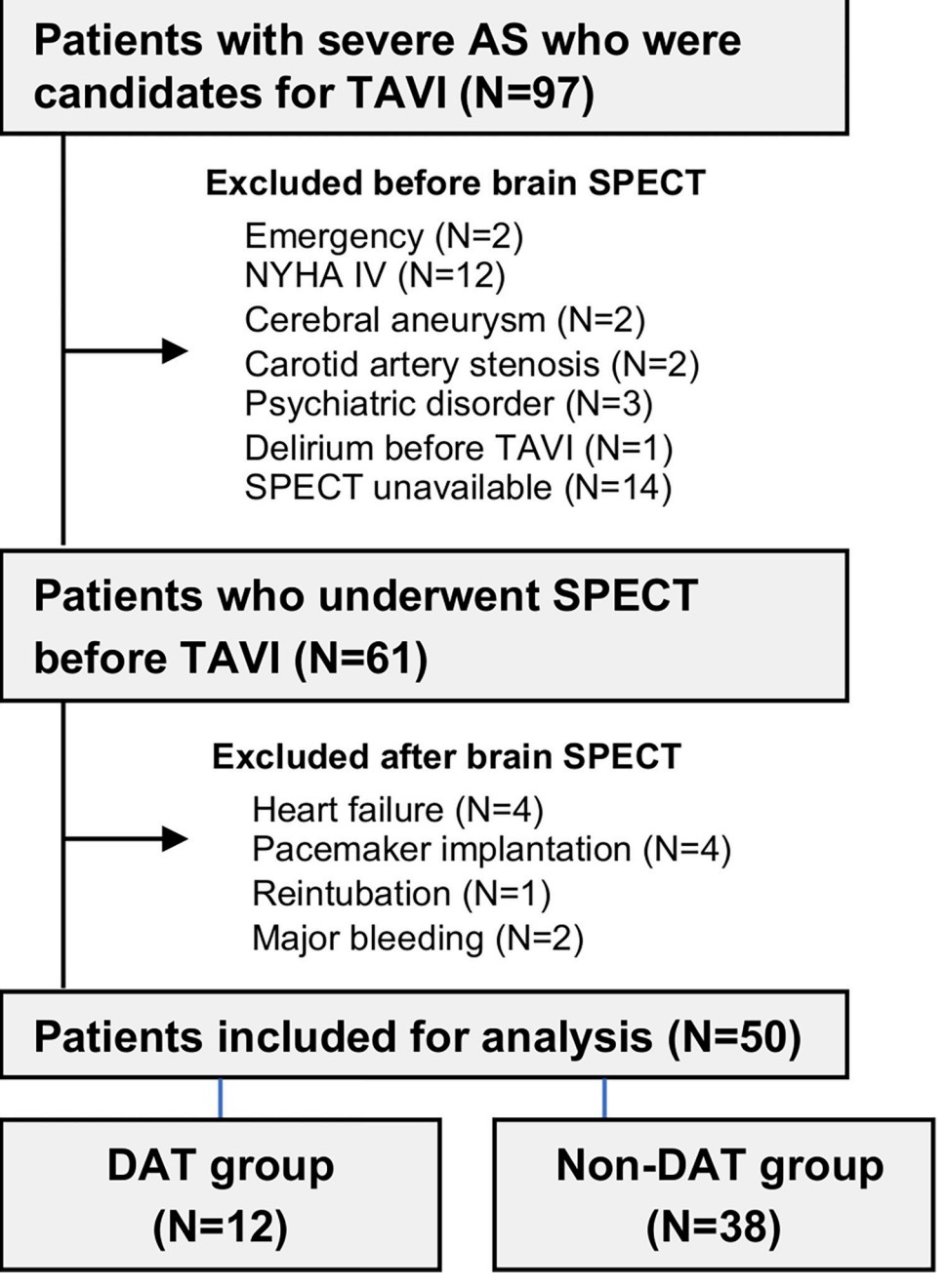

**Fig 1. Flow chart of patient enrollment in the present study.** Abbreviations: AS, aortic stenosis; DAT, delirium after TAVI; NYHA, New York Heart Association; SPECT, single-photon emission computed tomography; TAVI, transcatheter aortic valve implantation.

used. The study protocol was approved by the ethics committee of the Tohoku University Graduate School of Medicine (No. 2018-1-567) and was performed in compliance with the Declaration of Helsinki. Written informed consent was obtained from all patients.

## Baseline characteristics

We acquired preoperative information regarding age, sex, body mass index (BMI), NYHA classification, surgical risks (Society of Thoracic Surgeons score and Charlson comorbidity index) [24–26], presence of comorbidities, blood examination, echocardiography and pulmonary function tests. Frailty index was calculated from the following 5 items; weight loss, memory loss, non-exercise habits, fatigue from rest and walking speed decrease [27]. Each item was scored as 0 (robust), 1 to 2 (pre-frail) and 3 to 5 (frail). Cognitive and physical functions were investigated using mini mental state examination (MMSE) [28] and short physical performance battery (SPPB) [29, 30]. Furthermore, preoperative medication was investigated with regard to drugs reported to be associated with delirium [31]. We also assessed intraoperative (operative time, anesthesia time and the use of analgesics and sedatives) and postoperative factors (status of ambulation and the use of analgesics and sedatives, ICU stay).

## Assessment of delirium

Delirium was assessed daily after TAVI procedure by an anesthesiologist certified by the Japanese Society of Intensive Care Medicine using the confusion assessment methods for the intensive care unit (CAM-ICU) until delirium disappeared [32]. Then, a psychiatrist confirmed the diagnosis, subtype and duration of delirium according to the DSM-5 [33].

## SPECT image acquisition and pre-processing

Brain perfusion images were acquired with $^{99m}$Tc ethyl cysteinate dimer SPECT using a dual head gamma camera (Symbia E, Simens, Healthineers, Erlangen, Germany). A pre-processing of CBF images was conducted using SPM 12 as described previously [18]. Briefly, the images were normalized to the standard Montreal Neurological Institute space and were then smoothed with an isotropic Gaussian kernel by convolving a 12mm full-width at half maximum. These normalized and smoothed CBF images were used for the following neuroimaging analyses comparing the DAT group with the non-DAT group.

## Statistical analysis

We conducted statistical analyses using Stata statistical software version 17 (StataCorp) at a significance threshold of P<0.05 except for a whole brain voxel-wise analysis. Continuous variables were expressed as mean±standard deviation and were analyzed using the Student t-test. Categorical variables were expressed as n (%) and were analyzed using the Pearson $\chi^2$ statistic.

To test an association of brain perfusion before TAVI with development of DAT, we conducted the whole brain voxel-wise analysis comparing the pre-processed CBF images of the DAT group with those of the non-DAT group using SPM 12 as previously described [18]. The analysis used CBF of each voxel as independent variable, adjusted for age and sex. Results of the voxel-wise analysis were expressed at both exploratory threshold of P<0.001 uncorrected for multiple comparisons and more stringent threshold of P = 0.05 with family-wise error (FWE) corrections for multiple comparisons. The results from the voxel-wise analysis were validated using a region of interest (ROI)-based analysis with the easy-Z score imaging system (eZIS); the ratio of the number of voxels in the ROI with reduced CBF above z score >2 to all voxels in the ROI (decrease extent ratio; DE%) was compared between the two groups [34].

Usefulness of preoperative CBF for prediction of DAT was tested using the following regression analyses. We first performed univariable analyses with the presence of delirium as dependent variable and each of all survey items, including the result from the ROI-based analysis, as a dependent variables in the cohort with both of the DAT and non-DAT groups. Then, we performed multivariable logistic regression analyses with the presence of delirium as a dependent variable and combinations of the independent variables, which were significant in the univariable analyses (P<0.10), to obtain odds ratios, probabilities of significance, 95% confidence intervals and areas under the curve (AUC). Finally, based on the results of multivariable logistic regression analyses, ROC curves were generated to determine the best model to predict DAT.

## Results

### Patient characteristics

The development of DAT was in 12 patients (24%) (Fig 1). The preoperative characteristics of the DAT and the non-DAT groups as shown in Table 1. The DAT group had significantly higher frailty (frailty index, 3.58±1.38 vs. 2.45±1.16, P = 0.007), lower cognitive function (MMSE, 20.67±4.58 vs. 25.22±3.65, P = 0.007) and lower physical function (SPPB, 6.58±3.15 vs. 8.78±3.24, P = 0.025). There was no difference between the two groups in age, sex, BMI, NYHA class, Charlson comorbidity index, Society of Thoracic Surgeons score, comorbidities, blood tests or echocardiographic and spirometric parameters (P>0.05). Preoperative medications potentially associated with delirium were balanced between the two groups (P>0.05) (Table 2).

The intraoperative and postoperative findings are shown in Table 3. Subtypes of delirium were hypoactive (N = 8), hyperactive (N = 2) and mixed (N = 2) with duration of 2.67±1.84 days. The DAT group showed worse postoperative course relative to the non-DAT group: first-time oral intake (1.33±0.65 vs. 0.00, P = 0.002), first-time walking (3.67±2.50 vs. 2.29 ±1.81, P = 0.042) and discharge from ICU (1.92±1.38 vs. 1.18±0.73, P = 0.020) were delayed and short physical performance battery at discharge (5.92±3.26 vs. 8.74±3.70, P = 0.020) and return home rate (33.3% vs. 84.0%, P = 0.002) were in DAT group compared with the non-DAT group. There was no difference in durations of procedure and anesthesia or medications for sedation and analgesia between the two groups (P>0.05).

### Association of preoperative CBF with DAT

The whole-brain voxel-wise analysis showed that preoperative regional CBF in the insular cortex was lower in the DAT group than in the non-DAT group (P>0.05 FWE corrected) (Fig 2A–2C). This was supported by the ROI-based analysis; DE% of the insula was significantly greater in the DAT group than in the non-DAT group (17.6±11.5% vs. 7.0±11.3%, P = 0.007) (Fig 2D).

To specify associations of hypoactive delirium, the most common type of DAT, with cerebral perfusion, we have also compared CBF of patients with hypoactive delirium with that of non-DAT group. The results were similar to those comparing DAT group with non-DAT group: CBF in the insula was lower in patients with hypoactive delirium than in non-DAT group (P<0.05 FWE corrected) (S1 Fig). DE% of the insula in patients with hypoactive delirium was greater than the non-DAT group (16.6±9,8% vs. 7.0±11.3%, P = 0.03).

### Logistic regression analyses with DAT as an outcome

Univariable regression analyses showed that preoperative factors including frailty index, MMSE, SPPB and CBF in the insula predicted postoperative development of DAT (P<0.10)

**Table 1. Baseline patient characteristics.**

| | DAT group | Non-DAT group | P value |
|---|---|---|---|
| N | 12 (24%) | 38 (76%) | – |
| Age, years | 85.1±3.6 | 84.6±4.7 | 0.749 |
| Female | 11 (91.6) | 25 (65.8) | 0.140 |
| Body Mass Index, kg/m$^2$ | 22.9±2.7 | 22.2±2.4 | 0.388 |
| New York Heart Association functional class | | | |
| Class II | 2 (16.7) | 14 (36.8) | 0.292 |
| Class III | 10 (83.3) | 24 (63.2) | 0.292 |
| Society of Thoracic Surgeons score, % | 7.52±3.44 | 7.51±4.54 | 0.992 |
| Charlson comorbidity index, point | 2.67±1.30 | 2.76±1.48 | 0.841 |
| Short physical performance battery, point | 6.58±3.15 | 8.78±3.24 | 0.025 |
| Frailty index, point | 3.58±1.38 | 2.45±1.16 | 0.007 |
| Mini-mental state examination, point | 20.7±4.6 | 25.2±3.7 | 0.007 |
| Previous stroke/transient ischemic attack | 2 (16.7) | 6 (15.8) | 1.000 |
| Coronary artery disease | 3 (25.0) | 13 (34.2) | 0.668 |
| Hypertension | 8 (66.7) | 25 (65.8) | 1.000 |
| Diabetes mellitus | 2 (16.7) | 9 (23.7) | 1.000 |
| Cancer | 3 (23.1) | 7 (21.9) | 1.000 |
| Preoperative use of sleeping pills | 4 (33.3) | 8 (21.1) | 0.630 |
| Blood examination | | | |
| Hemoglobin, g/dL | 11.5±1.1 | 11.1±1.5 | 0.372 |
| Creatinine, mg/dL | 0.80±0.30 | 0.90±1.48 | 0.327 |
| Estimated glomerular filtration rate, ml/min/1.73 m$^2$ | 64.8±30.2 | 55.7±22.9 | 0.350 |
| Hemoglobin A1c, % | 5.80±0.74 | 6.06±0.78 | 0.305 |
| Brain natriuretic peptide, pg/mL | 506.1±441.2 | 456.7±701.6 | 0.774 |
| Echocardiography | | | |
| Left ventricular ejection fraction, % | 64.0±6.4 | 61.4±14.0 | 0.377 |
| E/e' | 22.8±10.9 | 19.6±9.2 | 0.404 |
| Aortic valve area, cm$^2$ | 0.61±0.12 | 0.69±0.26 | 0.180 |
| Peak velocity, m/s | 4.62±0.95 | 4.30±0.94 | 0.310 |
| Aortic mean gradient, mmHg | 51.8±21.2 | 45.5±21.1 | 0.379 |
| Low flow, Low gradient AS | 0 (0.0) | 3 (7.9) | 1.000 |
| Pulmonary function tests | | | |
| Vital capacity, L | 1.93±0.53 | 2.25±0.60 | 0.098 |
| Forced expiratory volume in one second, % | 77.5±12.1 | 96.5±25.1 | 0.723 |

Abbreviation: DAT, delirium after transcatheter aortic valve implantation

(Table 4). Of these, CBF in the insula had the highest AUC (0.818). An addition of CBF in the insula increased AUC of the model predicting DAT with each of frailty index, MMSE and SPPB (Fig 3). Although not achieving a significant difference from other models (P>0.05), the model including all of frailty index, MMSE, SPPB and CBF in the insula had the highest AUC (0.882) (Fig 3).

## Discussion

To support our hypothesis, the present study showed that preoperative CBF in the insula was significantly lower in patients with DAT than in those without it. Low CBF in the insula was raised as preoperative risk factor for DAT in addition to frailty, cognitive impairment and low

**Table 2. Preoperative drug use.**

|  | DAT group | Non-DAT group | P value |
|---|---|---|---|
| N | 12 (24%) | 38 (76%) | – |
| sleep medication, anti-anxiety drug | 3 (25.0) | 18 (47.4%) | 0.240 |
| antidepressant | 3 (25.0) | 2 (5.3) | 0.082 |
| atypical psychotropic drug | 2 (16.7) | 1 (2.6) | 0.139 |
| steroid | 1 (8.3) | 4 (10.5) | 1.000 |
| opioid | 1 (8.3) | 1 (2.6) | 0.426 |
| NSAIDs | 5 (41.7) | 11 (29.0) | 0.186 |
| benzodiazepines | 3 (25.0) | 15 (39.5) | 0.497 |
| digitalis | 1 (8.3) | 0 (0.0) | 0.240 |
| β-blocker | 6 (50.0) | 15 (39.5) | 0.738 |
| antiarrhythmic drug | 2 (16.7) | 12 (31.6) | 0.468 |
| oral cardiotonic drug | 0 (0.0) | 2 (5.3) | 1.000 |
| diuretic | 3 (25.0) | 11 (29.0) | 1.000 |

physical function. The addition of insular CBF to these traditional preoperative risk factors increased an accuracy of the model predicting DAT with frailty index, MMSE and SPPB (AUC 0.882). To the best of our knowledge, this is the first report demonstrating that preoperative CBF of the insula is associated with the onset of DAT and may be useful to predict DAT in elderly patients with severe AS.

## Association of CBF in the insula with DAT

In the present study, both whole brain voxel-wise and ROI-based analyses of the cerebral perfusion SPECT images provided the consistent results that preoperative CBF in the insula was

**Table 3. Intraoperative and postoperative variables.**

|  | DAT group | Non-DAT group | P value |
|---|---|---|---|
| N | 12 (24%) | 38 (76%) | – |
| Intraoperative findings |  |  |  |
| Duration of procedure, minute | 179.8±83.2 | 152.7±58.7 | 0.311 |
| Duration of anesthesia, minute | 304.8±22.3 | 282.1±12.5 | 0.412 |
| Medication |  |  |  |
| Pain medication | 4 (33.3) | 9 (23.7) | 0.707 |
| Sleep medication | 6 (50.0) | 8 (21.1) | 0.507 |
| Type of delirium |  |  |  |
| Hypoactive | 8 (66.6) | NA | – |
| Hyperactive | 2 (16.7) | NA | – |
| Mixed | 2 (16.7) | NA | – |
| Duration, day | 2.67±1.84 | NA | – |
| Postoperative course |  |  |  |
| First-time oral intake, day | 1.33±0.65 | 1.0 | 0.002 |
| First-time walking, day | 3.67±2.50 | 2.29±1.81 | 0.042 |
| Stay at intensive care unit, day | 1.92±1.38 | 1.18±0.73 | 0.020 |
| Short physical performance battery at discharge | 5.92±3.26 | 8.74±3.70 | 0.020 |
| Return home rate | 4 (33.3) | 32 (84.0) | 0.002 |

Abbreviation: DAT, delirium after transcatheter aortic valve implantation.

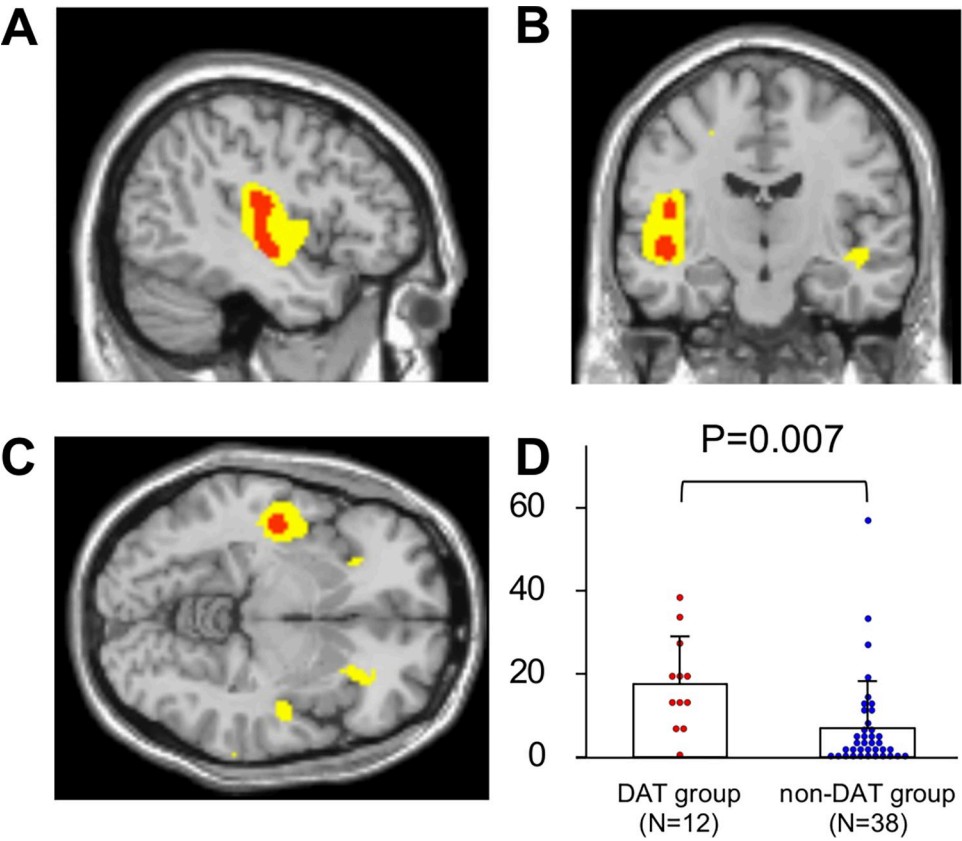

**Fig 2. Lower insular perfusion in the DAT group compared with the non-DAT group.** Results of the whole brain voxel-wise analysis shown in the sagittal (A), axial (B), and the coronal slices (C) (red areas, P<0.05 with family-wise error corrections; yellow areas, P<0.001 without corrections for multiple comparisons). In the region of interest-based analysis, decrease extent ratios were higher in the DAT group (N = 12) than in the non-DAT group (N = 38) (D). Abbreviations: DAT, delirium after transcatheter aortic valve implantation.

lower in the DAT group compared with the non-DAT group. CBF in the insula was lower also in patients with hypoactive delirium than the non-DAT group. Previous SPECT studies focus mainly on CBF after the onset of delirium, when CBF was reduced in several cortical and sub-cortical structures [22, 23]. Functional MRI assessments before elective surgery showed that reduced connectivity strength and efficiency of structural brain networks were found in risk groups for delirium [35]. These findings indicate that brain dysfunction is associated with delirium.

The insula has neural connections with multiple brain regions, including the frontal lobe, parietal lobe, temporal lobe and limbic system (hippocampus, cingulate gyrus and amygdala)

**Table 4. Results of univariable regression analyses with delirium as an outcome.**

|  | Odds ratio | 95% confidence interval | P value | Area under curve |
|---|---|---|---|---|
| Short physical performance battery | 0.855 | (0.711, 1.028) | 0.096 | 0.701 |
| Mini mental state examination | 0.757 | (0.625, 0.916) | 0.004 | 0.787 |
| Frail index | 2.037 | (1.168, 3.552) | 0.012 | 0.734 |
| Cerebral blood flow in the insula | 1.071 | (1.011, 1.135) | 0.021 | 0.818 |

Only the results with P<0.10 are shown. Cerebral blood flow in the insula was expressed as decrease extent ratio from the region of interest-based analysis.

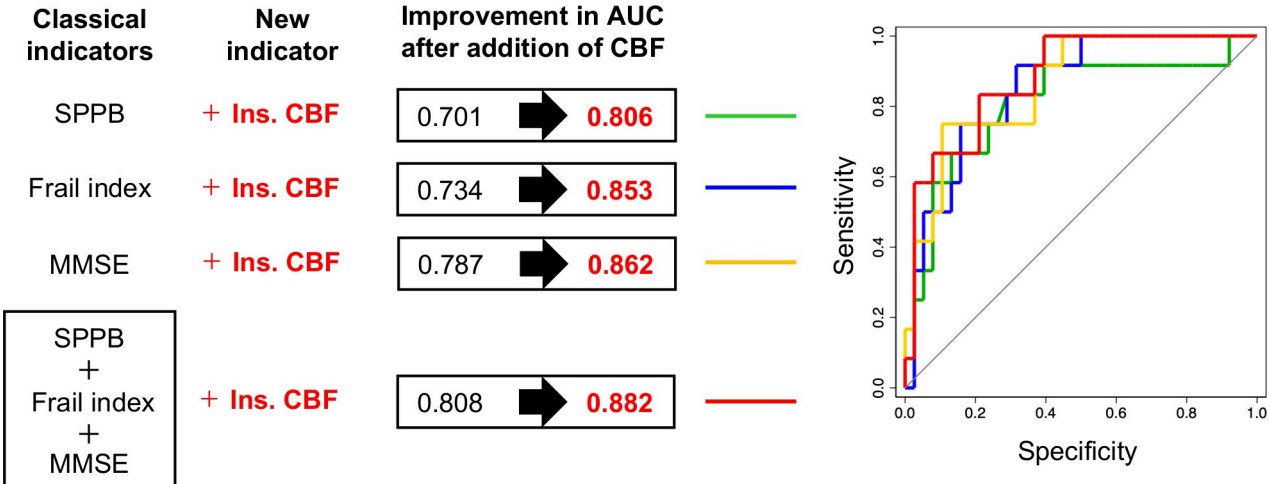

**Fig 3. Results of multivariable logistic regression analyses with an addition of CBF in the insula to classical indicators.** Abbreviations: AUC, area under curve; CBF, cerebral blood flow; Ins, insular; MMSE, mini mental state examination; SPPB, short physical performance battery.

[36]. Functional differences of the anterior, middle and posterior insula [37–39] and those of the right and left insula [38, 40, 41] have been reported. The anterior insula is associated with attention, awareness, emotion, pain and negative affect, the middle insula with auditory to vocal conversion and the posterior insula with auditory processing of language and somato-sensation [37–39]. Disturbance of these functions seems to be similar with core symptoms of delirium [33], suggesting an association of dysfunction in the insula with DAT. The left insular is associated with activity of the parasympathetic nervous system, while sympathetic arousal occurs in stimulation of the right insular [38, 41]. Patients with right insular lesions show delirium-like behaviors compared with those with non-insular or left insular lesions [40]. In the present study, however, CBF in the bilateral insula was lower in patients with hypoactive delirium compared to the non-DAT group. The DAT group, including all of the three types of delirium, also showed a similar tendency of lower CBF in the bilateral insula than the non-DAT group. The laterality of brain abnormalities associated with DAT should be investigated in the future studies.

## CBF in the insula as direct factor for DAT onset

Several preoperative factors for the postoperative development of delirium have been reported [6, 42]. Most of them are related to comorbidity or risk assessment indicators [6, 43]. In the present study, CBF in the insula before TAVI had higher AUC (0.818) to predict DAT compared with preoperative frailty, cognitive decline and physical decline. The addition of CBF to these three factors substantially improved the AUC to 0.882. There are three factors that contribute to the onset of delirium: direct factors, preparatory factors and triggering factors [9]. Organ damage, including the reduction of CBF, is classified as direct factor while frailty, cognitive decline and physical decline are regarded as preparatory factor. The results of the present study support the concept that combined effects of the three factors, including hypoperfusion of the insula, lead to the onset of DAT (Fig 4).

The incidence of DAT in the present study (24.0%) was consistent with those of the previous studies using the same assessment battery (21–44%) [8, 44]. The DAT group showed worse postoperative course, including delayed first-time oral intake, first-time walking and discharge from ICU and lower physical performance at discharge and return home rate,

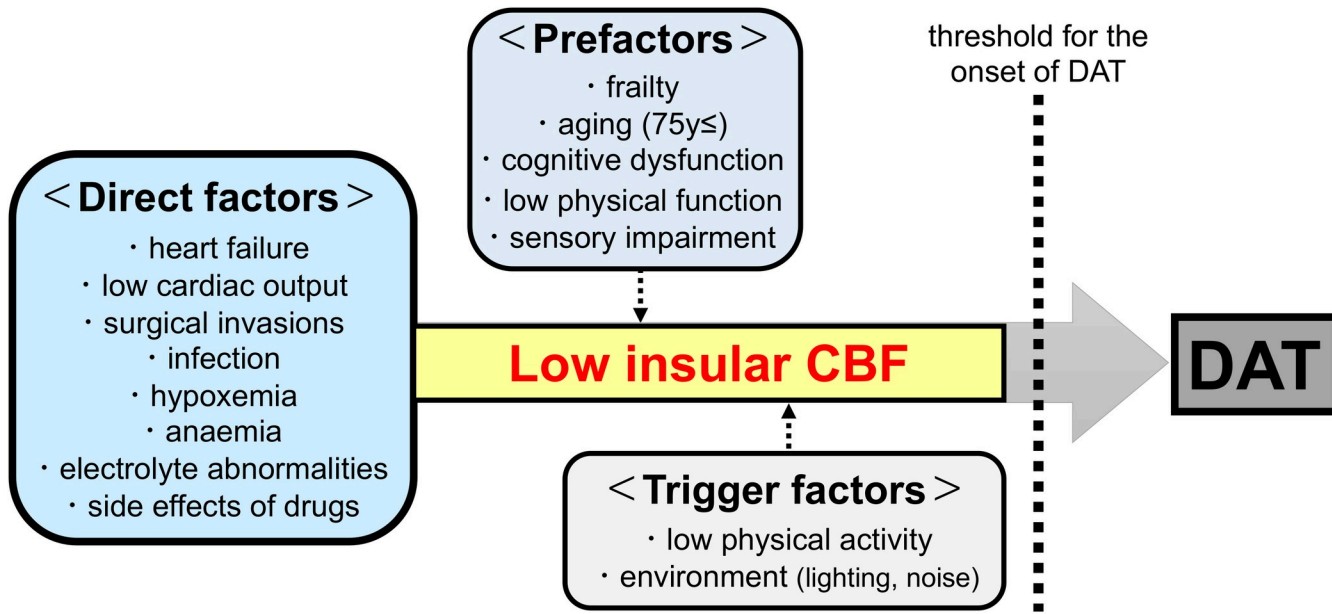

**Fig 4. Multifactorial nature of DAT and possible involvement of preoperative CBF in the insula.** Significant (P>0.10) factors that contribute the onset of DAT in the present study are emphasized as red-front. Abbreviations: CBF, cerebral blood flow; DAT, delirium after transcatheter aortic valve implantation.

relative to the non-DAT group. Assessment of CBF in the insula may detect patients with high risk of DAT preoperatively. Moreover, CBF in the insula is increased by exercise and meditation [45, 46], suggesting that non-pharmacological approaches may prevent the onset of DAT through improvement of CBF reduction. These findings indicate that preoperative CBF in the insula may be a useful therapeutic target for DAT.

## Study limitations

Several limitations should be raised for the present study. Firstly, the present study is a single-center study with a small number of patients. A study with a larger number of patients is needed. Secondly, the present study did not have a control group without TAVI although this issue is difficult to address for ethical reason. Thirdly, the present study did not test other modalities of brain structure and function. Analyses of brain structural MRI [47], functional MRI [35] and positron emission tomography [48] could be considered in the future. Fourthly, although TAVI has been increasingly performed under local anesthesia, all TAVI procedures were performed under general anesthesia in the present study. The impact of TAVI under local anesthesia remains to be addressed.

## Conclusions

In the present study, we were able to demonstrate for the first time that reduced preoperative CBF in the insula assessed by SPECT is a useful predictor for postoperative development of delirium in patients with severe AS undergoing TAVI procedure.

## Supporting information

**S1 Fig. Lower insular perfusion in patients with hypoactive delirium compared with the non-DAT group.** Results of the whole brain voxel-wise analysis at a significance threshold of P<0.05 with family-wise error corrections (red regions) and P<0.001 without multiple

comparisons (yellow regions).
(TIF)

## Acknowledgments

We thank Yuko Ogata, an anesthesiologist in Tohoku University Hospital, for assessment of delirium in the present study.

## Author Contributions

**Conceptualization:** Masashi Takeuchi, Hideaki Suzuki, Yasuharu Matsumoto.

**Data curation:** Masashi Takeuchi, Hideaki Suzuki.

**Formal analysis:** Masashi Takeuchi, Hideaki Suzuki.

**Funding acquisition:** Hideaki Suzuki.

**Investigation:** Masashi Takeuchi, Hideaki Suzuki, Yasuharu Matsumoto, Yoku Kikuchi, Toshihiro Wagatsuma, Jun Sugisawa, Satoshi Tsuchiya, Kensuke Nishimiya, Kiyotaka Hao, Shigeo Godo, Tomohiko Shindo, Takashi Shiroto, Jun Takahashi, Kiichiro Kumagai.

**Methodology:** Masashi Takeuchi, Hideaki Suzuki, Kentaro Takanami.

**Project administration:** Hideaki Suzuki, Yasuharu Matsumoto, Satoshi Yasuda.

**Resources:** Masashi Takeuchi, Hideaki Suzuki, Yasuharu Matsumoto, Yoku Kikuchi, Toshihiro Wagatsuma, Jun Sugisawa, Satoshi Tsuchiya, Kensuke Nishimiya, Kiyotaka Hao, Shigeo Godo, Tomohiko Shindo, Takashi Shiroto, Jun Takahashi, Kiichiro Kumagai.

**Software:** Masashi Takeuchi, Hideaki Suzuki, Kentaro Takanami.

**Supervision:** Hideaki Suzuki, Yasuharu Matsumoto, Masahiro Kohzuki, Kei Takase, Yoshikatsu Saiki, Satoshi Yasuda, Hiroaki Shimokawa.

**Validation:** Kentaro Takanami.

**Writing – original draft:** Masashi Takeuchi, Hideaki Suzuki.

**Writing – review & editing:** Yasuharu Matsumoto, Yoku Kikuchi, Kentaro Takanami, Toshihiro Wagatsuma, Jun Sugisawa, Satoshi Tsuchiya, Kensuke Nishimiya, Kiyotaka Hao, Shigeo Godo, Tomohiko Shindo, Takashi Shiroto, Jun Takahashi, Kiichiro Kumagai, Masahiro Kohzuki, Kei Takase, Yoshikatsu Saiki, Satoshi Yasuda, Hiroaki Shimokawa.

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
