## [Decision Letter · Decision Letter 0]

29 Jun 2022

PONE-D-22-15512Prediction of the development of delirium after transcatheter aortic valve implantation using preoperative brain perfusion SPECTPLOS ONE

Dear Dr. Yasuda,

Thank you for submitting your manuscript to PLOS ONE. After careful consideration, we feel that it has merit but does not fully meet PLOS ONE’s publication criteria as it currently stands. Therefore, we invite you to submit a revised version of the manuscript that addresses the points raised during the review process.

ACADEMIC EDITOR: The Reviewer's comments suggested major revisions

We look forward to receiving your revised manuscript.

Kind regards,

Antonino Salvatore Rubino, M.D., Ph.D.

Academic Editor

PLOS ONE

Journal Requirements:

Additional Editor Comments:

The Reviewer's comments suggested major revisions

Reviewers' comments:

Reviewer's Responses to Questions

**Comments to the Author**

1. Is the manuscript technically sound, and do the data support the conclusions?

Reviewer #1: Yes

2. Has the statistical analysis been performed appropriately and rigorously? 

Reviewer #1: Yes

3. Have the authors made all data underlying the findings in their manuscript fully available?

Reviewer #1: Yes

4. Is the manuscript presented in an intelligible fashion and written in standard English?

Reviewer #1: Yes

5. Review Comments to the Author

Reviewer #1: In this manuscript by Takeuchi et al., “Prediction of the development of delirium after transcatheter aortic valve implantation using preoperative brain perfusion SPECT” was shown. This is very excellent study. Some comments were there.

#1: Hypoactive delirium

Most of the DAT might be hypoactive delirium. The type of the delirium should be explained more specifically through the manuscript.

#2: CBF in the right insular cortex

Reduced CBF in the right insular cortex seems to be associated with DAT. Please specify this through the manuscript. Especially, in the Table 4, the relationship between CBF in the right insular cortex and the delirium was not clear.

#3: Diuretic use

Please provide the information for the diuretic use, which might be associated with the study results.

#4: Low EF/low gradient

In this study population, the EF seems to be preserved. Might be there some patients with AS with low EF and/or low gradient? Some patients were assessed severe AS using dobutamine?

#5: Diastolic function

Diastolic function might be associated with delirium. Please provide the information such as the E/e’.

#6: Regional cerebral blood flow

It would be helpful if the specific description for CBF was provided. In the Fig3 and Fig4, CBF sounds like global CBF. Thus, use of the term such as “the reduced CBF in the right insular cortex” would be helpful.

#7: Discussion for the relationship between reduced CBF in the right insular cortex and delirium

It would be clear if the relationship between reduced CBF in the right insular cortex and delirium was discussed more specifically. The main symptoms were anergia or hypoactivity like depressive symptom?

#8: Symptom change

It is interesting if the change of the symptom such as chest pain was provided. In the delirium patients, the chest pain

#9: Stroke during TAVI procedure

Was there any patients with stroke during TAVI procedure?

6. PLOS authors have the option to publish the peer review history of their article (what does this mean?). If published, this will include your full peer review and any attached files.

Reviewer #1: **Yes: **Michiaki　Nagai

---

## [Author Response · Author response to Decision Letter 0]

23 Aug 2022

We appreciate the favorable comments from the editor and reviewer. Please see our cover letter and responses to reviewer documents with respect to specific reviewer and editor comments. We hope our revised manuscript may again be considered for publication in the PLOS ONE.

---

## [Editor Report · Decision Letter 1]

7 Oct 2022

Prediction of the development of delirium after transcatheter aortic valve implantation using preoperative brain perfusion SPECT

PONE-D-22-15512R1

Dear Dr. Yasuda,

We’re pleased to inform you that your manuscript has been judged scientifically suitable for publication and will be formally accepted for publication once it meets all outstanding technical requirements.

Kind regards,

Antonino Salvatore Rubino, M.D., Ph.D.

Academic Editor

PLOS ONE

Additional Editor Comments (optional):

The authors satisfactorily addressed all Reviewer's comments
---

## [Editor Report · Acceptance letter]

26 Oct 2022

PONE-D-22-15512R1 

Prediction of the development of delirium after transcatheter aortic valve implantation using preoperative brain perfusion SPECT 

Dear Dr. Yasuda:

I'm pleased to inform you that your manuscript has been deemed suitable for publication in PLOS ONE. Congratulations! Your manuscript is now with our production department. 

Kind regards, 

on behalf of

Dr. Antonino Salvatore Rubino 

Academic Editor

PLOS ONE